# Influence of a Child’s Cancer on the Functioning of Their Family

**DOI:** 10.3390/children8070592

**Published:** 2021-07-13

**Authors:** Anna Lewandowska

**Affiliations:** Institute of Healthcare, State School of Technology and Economics, 37-500 Jaroslaw, Poland; am.lewandowska@poczta.fm; Tel.: +48-698757926

**Keywords:** cancer, problems, child, parents, family problems

## Abstract

Background—A child’s cancer affects their entire family and is a source of chronic stress for a sick child, as well as for their parents and siblings. It deprives them of the feeling of security; introduces uncertainty, fear and anxiety; and destabilises their life. It mobilises the family since they have to reconcile the treatment and frequent appointments at the hospital with the hardships of everyday life. The emotional burden they have to deal with is enormous. Recognition of the needs of such a family allows for the implementation of support, psychosocial care and psychoeducation, as well as the provision of reliable information. Patients and Methods—A population survey was conducted between 2015 and 2020. Caregivers of children diagnosed with cancer were invited to participate in the study to assess their problems and needs. Results—All respondents in their legal status were parents of children with cancer. The study included 800 people, where women accounted for 85% and men accounted for 15%. The mean age of the mother was 38.09, SD = 7.25, and the mean age of the father was 41.11, SD = 7.03. The occurrence of problems negatively correlated with both the age of the parents (*p* < 0.0001) and the level of education (*p* < 0.0001). Parents who admitted having financial problems more often reported problems of a different kind; moreover, financial problems were more often reported by parents of children who were ill for a longer time (*p* = 0.01). Conclusions—Parents of children suffering from cancer reported numerous psychological, social and somatic problems. The identification of problems through screening should translate into specific interventions, thus creating support for the families of children with cancer. Promoting coping with difficult emotions and the ability to solve problems when a child is ill has a positive effect on the functioning of the family.

## 1. Introduction

A child’s cancer affects their entire family and is a source of chronic stress for a sick child, as well as for their parents and siblings. It deprives them of the feeling of security; introduces uncertainty, fear and anxiety; and destabilises their life. It mobilises the family since they have to reconcile the treatment and frequent appointments at the hospital with the hardships of everyday life. The emotional burden they have to deal with is enormous. Families are burdened with thoughts of the irreversibility and duration of the disease and the fact that the child will have to cope with physical and mental pain. Goals, priorities, values and plans for the near and further future are changing. The family learns to function in new circumstances, cope with hardships and experience difficult emotions and conflicts. They have to reconcile the requirements of caring for a sick child with professional work, financial concerns, caring for healthy siblings and contact with the social environment. This makes the process of treating cancer difficult, burdensome and sometimes even unacceptable for the whole family [1,2,3].

The family experiences various emotional phases of adapting to the disease. The intensity and kind of experience depend on the personality and mutual relations in the family. There are stages of anger, resentment and pain, which are preparatory stages for parting with the sick person. Each stage causes changes in the behaviour and relationships between family members, communication problems, somatic problems and role swaps. Among caregivers, traumatic symptoms may develop and persist throughout the first year of therapy. Taking into account the psychological problems faced by the patient’s family, it is also important to consider how siblings react to the disease. Taking up a difficult topic, such as an illness, is often unbearable for a parent who wants to protect their child from pain at all costs. Children, however, are very sensitive to hidden anxieties or emotions. Through noticing changes in everyday behaviour, the rhythm of the day and appearance, they experience a crisis caused by an illness in solitude. The atmosphere of constant secrets and hidden emotions has a negative effect on the child, influencing their development. Secretly experienced emotions are reflected in their behaviour, causing excessive tearfulness, sleep disturbances and aggressive behaviour. When a child learns about a sibling’s illness, they may feel anger, regret, rejection, guilt and responsibility for their illness, and even fear that the parent may become ill [4,5,6].

How the family copes with these demands determines its adaptation to the disease and treatment. It is assumed that most families of a child diagnosed with cancer experience stress but can cope with the disease and its treatment and adapt to this situation. However, there are families that, due to individual, family, social and economic factors, are exposed to more difficulties in the case of a child’s cancer and have to overcome more adversities than other families. These families experience more severe stress and therefore need more attention, support and professional interventions. Psychosocial care for the patient and their family is extremely important. It turns out to be an inseparable element of comprehensive care that should be covered by every family in which a child has been diagnosed with cancer [1,7,8].

The assessment of the impact of a child’s cancer on the functioning of their relatives and the analysis of the problems and needs of parents and guardians of the child in all areas of life is necessary for planning and conducting holistic care. Recognising the problems of such a family is crucial, as it allows for the implementation of support, psychosocial care, psychoeducation and the provision of reliable information. Moreover, the assessment of the problems, needs and concerns of the patient’s family is an important role for health professionals, as clear guidelines and standards for assessing psychosocial needs and their effective assessment provide knowledge about the type of interventions that should be implemented by a multidisciplinary team taking care of the patient and their family.

## 2. Objective of this Research

This study aimed to assess the impact of a child’s cancer on the functioning of their relatives and to analyse the problems and needs of parents and guardians of the child in all areas of life. Assessing problems provides an opportunity to better understand the experiences of families, is essential in prioritising resource allocation and it provides healthcare professionals a broader view of the family of a child with cancer.

## 3. Material and Method

### 3.1. Study Design

A qualitative questionnaire study was conducted in paediatric oncology and haematology departments of clinical hospitals and oncology clinics in the Podkarpackie Province in 2015–2020. The clinical hospitals and clinics provided care for paediatric patients living in the Podkarpackie Province that were diagnosed with cancer. All patients were treated and cared for by the public health system. Caregivers of children diagnosed with cancer were invited to participate in the study to assess their problems and needs. Due to the small size of the sample, the share of respondents with fairly consistent characteristics was important. The inclusion criteria were: confirmed diagnosis of childhood cancer in a child under 18 years of age, without previous chronic or life-threatening diseases and understood the Polish language. The exclusion criterion was the diagnosis of a neoplastic disease lasting shorter than three months, as the initial period of diagnosis is associated with huge psychological stress and the need to adapt not only by the patient but the entire family, which may introduce errors in the results. Caregivers in very poor physical and emotional conditions were excluded from the study. 

### 3.2. Participants

Parents or legal guardians of children were invited to participate in the study during their outpatient visit or during the child’s hospitalisation. Only the parents who performed the main care role in each family took part in the study, i.e., was present during the stay in the hospital or during the inspection at the clinic. Each invited person was informed about the purpose of the study. After obtaining informed consent, subjects were asked to complete a questionnaire. The respondent was allowed to complete an online questionnaire or a paper version.

### 3.3. Research Procedures

The study was approved by the Bioethics Committee (resolution no. 386/2009 and 12 April 2017). Participation in this study was voluntary and anonymous, and respondents were informed of their right to refuse or withdraw from the study at any time. Each participant was informed about the purpose of the study and the time of completion of the study. Families were invited to participate during the child’s hospitalisation or an outpatient visit. After giving informed consent, the parents completed the questionnaire.

### 3.4. Method

#### 3.4.1. Questionnaire

The method used in the research was a qualitative, direct, individual and structured interview, which was in-depth and focused. The qualitative interview questionnaire was a standardised measuring instrument. The questionnaire was verified by testing a group of 30 guardians over a month and was assessed for internal consistency (yielding a Cronbach’s alpha of 0.83). Parents completed a questionnaire that included open-ended, single and multiple-choice questions to obtain record and epidemiological information, as well as to assess financial, caring, psychological, somatic, communication and family problems.

#### 3.4.2. Scales

##### Zung’s Self-Assessment Scale

A scale called the Zung’s Self-Rating Depression Scale was used to assess the well-being of the participants, the severity of depression symptoms. In this test, the respondent marks the answers to 20 questions about their well-being, choosing one of four answers on a four-point scale from 1 to 4 points, where 1 means no or for a short time, and 4 for most or all of the time. The sum of points for all answers can range from 20 to 80, but the higher the score, the worse the well-being and the deeper the depressive state. For this scale, Cronbach’s coefficient was 0.95.

##### Dysfunction Assessment Scale

The scale for assessing dysfunction due to biopsychosocial problems is a screening tool that was filled in by parents or guardians of a sick child. It contains a visual analogue scale to assess the functioning of the family in terms of the following problems: financial difficulties, lack of social support, lack of family support, psychological problems, problems in the relationship between partners, problems with substance abuse, health problems, legal problems, care problems towards the patient child and care problems towards siblings. All items are rated on a scale of 0 to 3, with 0 meaning no problem, 1 meaning some of the time, 2 meaning most of the time and 3 meaning all the time. People declaring problems with a score between 0–10 have a minimal risk of dysfunction, 11–20 means a moderate risk and 21–30 means a high risk. For this scale, Cronbach’s coefficient was 0.94.

### 3.5. Data Analysis

The analysis used descriptive statistics and confidence intervals in the assessment of the participants’ characteristics, metric and demographic data and the assessment of problems. Statistical characteristics of continuous variables are presented in the form of arithmetic means, standard deviations and medians. Statistical characteristics of step and qualitative variables are presented in the form of numerical and percentage distributions after using Student’s t-test or the Mann–Whitney U test. Correlations were determined using Pearson’s test, while χ^2^ was used for comparisons between the groups. Significance was assessed at the level of *p* < 0.05. The repeatability of answers to individual questions was assessed using Cohen’s kappa statistics. Missing data were excluded from all analyses. 

## 4. Results

### 4.1. Demographic Data

All respondents in their legal status were parents of children with cancer. A total of 800 parents completed the questionnaire and 800 people were included in the survey, where women accounted for 85% and men accounted for 15%. The mean age of the mother was 38.2, SD = 7.25, and the mean age of the father was 41.1, SD = 7.03. More than half of the surveyed women declared achieving higher education (52%, 95% CI: 51–54), while 32% achieved secondary education (95% CI: 30–35) and 16% achieved vocational education (95% CI: 12–19). In the case of men, those with secondary education predominated (48%, 95% CI: 41–50), 31% (95% CI: 30–33) declared they achieved vocational education, and higher education was achieved by 21% (95% CI: 17–24). Most parents lived in the city (68%, 95% CI: 62–70), while the rest were rural residents (32%, 95% CI: 29–35). Most were married (74%, 95% CI: 71–77), while 12% (95% CI: 10–15) were single parents and 11% (95% CI: 9–13) were divorced. The majority of care for children was provided by the mothers (87%, 95% CI: 84–89). The diagnosed cancers among the children were: leukemias (54%, 95% CI: 51–57), brain tumors (19%, 95% CI: 14–20) and solid tumors (27%, 95% CI: 26–30). Most respondents had one child (45%, 95% CI: 44–50), while 41% (95% CI: 38–42) had two children, 10% had three children (95% CI: 5–10), and 4% had four children (95% CI: 2–5). The median ages of the first, second, third and fourth children were 11, 8, 6 and 3 years, respectively. The occurrence of problems negatively correlated with both the age of the parents (mean score, 31 versus 49, *p* < 0.0001) and the level of education (χ^2^ = 0, *p* < 0.0001). Parents with higher education and who were under the age of 50 coped better with the child’s disease. Other descriptive statistics identifying the studied group are presented in Table 1.

### 4.2. Financial Problems

More than half of the families (68%, 95% CI: 64–70) described their financial situation as average and 13% (95% CI: 11–15) described it as very bad. Almost half of the surveyed families (44%, 95% CI: 41–45) believed that their financial situation worsened with the child’s disease to a moderate extent, 39% (95% CI: 38–40) believed that it worsened to a large extent, 13% believed that it worsened to a small extent (95% CI: 11–15) and 4% (95% CI: 3–5) of families believed that their financial situation had not changed. For 26% (95% CI: 24–27) of the families, the child was an additional financial burden. In 76% (95% CI: 74–78) of families, the source of income was professional work, and 24% (95% CI: 21–25) had resigned from work. According to the results of a simple analysis of the level of problems, there was a statistically significant relationship with the financial situation. Parents who admitted having financial problems reported having problems of a different kind more often; moreover, financial problems were more often reported by parents of children who were ill for a longer time (*p* = 0.01) (Table 1).

### 4.3. Caring Problems

Most children were independent (70%, 95% CI: 69–75), with only 30% (95% CI: 28–35) needing parental help. The largest number of children, as much as 43% (95% CI: 41–45), required moderate help, while 39% (95% CI: 37–40) needed a large amount of help and 18% (95% CI: 15–20) of children needed little help. Among the parents, 47% (95% CI: 44–50) of them believed that their children have somatic problems, 55% (95% CI: 53–57) had psychological problems and 61% (95% CI: 41–50) had social problems (Table 2). A total of 63% (95% CI: 61–65) of parents did not know whether their child would be independent in the future, 28% (95% CI: 27–30) believed that their child will be independent and only 9% (95% CI: 4–10) families believed that their child would not become independent. Most caregivers adjusted their family life to care for their sick child, 14% (95% CI: 11–17) gave up free time, 10% (95% CI: 6–12) adjusted their working hours and 24% (95% CI: 21–27) had to quit all work. Parents found it hardest to take care of their children because of the lack of state support (57%, 95% CI: 53–59) and their difficult financial situation (22%, 95% CI: 20–25) (Figure 1). In the studied group, there was a very strong positive linear relationship between the caring problems and the professional work of the caregivers (+0.993). This means that professionally active people reported problems more often in their sick children. Parents of children who were ill for longer periods reported care problems less frequently. The differences were statistically significant (*p* = 0.01).

### 4.4. Psychological Problems

After receiving the information that the child was sick, 20% (95% CI: 19–22) of parents were devastated, 35% (95% CI: 31–37) denied the disease and 45% (95% CI: 44–46) did not believe the diagnosis. At the time of the survey, parents felt anxiety (75%, 95% CI: 71–78), resignation (10%, 95% CI: 4–12), helplessness (10%, 95% CI: 4–12) about the child’s future and determined to ensure proper care (7%, 95% CI: 4–10), while 15% (95% CI: 11–20) had had not come to terms with the child’s disease. Among the parents, only 2% (95% CI: 1–3) blamed themselves for their child’s sickness, while the majority (98%, 95% CI: 94–99) did not see the fault on their side. Parents received the most support from their own families (35%, 95% CI: 31–38) (Figure 2). As many as 36% (95% CI: 34–40) of families encountered insensitivity or avoidance by friends, as well as a lack of community support (22%, 95% CI: 21–24). A total of 98% (95% CI: 94–99) of parents believed that they had good contact with their sick child. Overall, 25% (95% CI: 21–27) of parents benefited from psychiatric care due to various problems in the psychological sphere (Figure 3). Regarding the screening of the occurrence of mental disorders among parents, after using Zung’s Depression Self-Assessment Scale (SDS) and Self-Assessment Anxiety Scale (SAS), an index of 50 or more in the SDS (indicating depression) was reported by 15% (95% CI: 11–19) of respondents and a score of 45 and above in the SAS (indicating anxiety) was reported by 77% (95% CI: 74–79) of the respondents. Of the total sample, 11% (95% CI: 8–15) were taking psychotropic drugs because of their diagnosis of depression. A feeling of exhaustion was reported by 53% (95% CI: 51–55) of mothers of sick children. In the studied group, there was a very strong positive linear relationship between the problems of the psychological sphere and the number of children and place of residence (+0.993). The results were significantly higher for families with more than one child and those living in rural areas.

### 4.5. Somatic Problems

The child’s illness also affected the parents’ physical well-being and ailments. The most frequently reported symptoms from the symptom list were fatigue (68%, 95% CI: 64–70) and difficulty sleeping (51%, 95% CI: 49–53) (Table 3). According to the results of a simple analysis of the frequency and type of ailments, there was a statistically significant relationship with the duration of illness. Parents of children who were ill for more than 3 years reported somatic disorders more often. The differences were statistically significant (*p* = 0.03).

### 4.6. Communication Problems

Most parents were informed about the child’s illness by a doctor (87%, 95% CI: 84–89), while in other cases by a nurse (6%, 95% CI: 4–9) or a psychologist (7%, 95% CI: 4–9). The sources of knowledge about the child’s disease were conversations with other parents (17%, 95% CI: 14–20), associations and foundations (19%, 95% CI: 14–20), doctors (12%, 95% CI: 10–15), professional literature (8%, 95% CI: 7–10) and nurses (5%, 95% CI: 4–6). As many as 58% (95% CI: 55–59) of the respondents assessed their knowledge as sufficient, 30% (95% CI: 29–32) as partly sufficient, and 12% (95% CI: 11–15) as insufficient. Problems with the flow and obtaining information about their sick child from healthcare professionals were reported by as many as 79% (95% CI: 74–80) of parents. The most common shortcomings were: the treatment plan (19%, 95% CI: 14–20) and prognosis (78%, 95% CI: 74–80).

### 4.7. Family Problems

In most cases, the child’s disease did not change the relationship within the family (41%, 95% CI: 39–43), it strengthened family ties in 32% (95% CI: 30–35) and the family relations deteriorated in 27% (95% CI: 25–29). According to the parents, more than half of the siblings (57%, 95% CI: 54–59) had a very good attitude towards the sick child; however, 43% (95% CI: 41–45) had minor or major conflicts. Problems with other children were reported by 68% (95% CI: 66–70) of the surveyed parents. The mentioned problems in siblings were tantrums (48%, 95% CI: 43–50), learning difficulties (21%, 95% CI: 19–23) and behavioural problems (18%, 95% CI: 15–23). As many as 47% (95% CI: 45–49) of parents admitted that they had problems with balancing the needs of the whole family, mainly pointing to the lack of time for the remaining children (61%, 95% CI: 60–63) and problems with the organisation of everyday life (43%, 95% CI: 41–45). Family conflicts were reported by 27% (95% CI: 25–29) of the respondents (Figure 4).

### 4.8. Occurrence of Dysfunction

The obtained results indicated which group of risk of biopsychosocial problems the families belonged to. A total of 34% of families were at the lowest, minimum level of risk; 46% at a higher, moderate level of risk; and 20% of the respondents had a high level of risk of developing psychosocial problems (Table 4).

## 5. Discussion

This study looked at the impact of a child’s cancer on the functioning of their family. Diagnosing a child’s cancer, a potentially fatal disease, is one of the most life-changing and stressful experiences a parent and immediate family may encounter. Parents and family relationships are undoubtedly influenced by the treatment process, side effects, financial burden and, above all, the fear of death. The child’s illness forces the parents to modify their current roles, acquire new skills and satisfy the new needs of the child. This difficult situation may theoretically have negative consequences for the child’s parents, siblings and family relations, but such an experience may also strengthen the family [1,2,3,7,8].

This study analysed the structure of families, where it was shown that parental responsibilities were unevenly distributed and that mothers more often played the main role of caring for a sick child (87%). A similar situation was confirmed by other researchers, also in countries where equality of duties is the main model. The role includes spending time in the hospital, administering medications and providing care for health problems [2,3,4,5,6,7,8,9,10,11].

A child’s illness has a significant impact on the parents’ professional life and finances. The study showed that 96% of parents felt the financial burden related to treatment, rehabilitation and care to a greater or lesser extent, and 24% of them had to completely give up their work. Lau et al. showed that during the active treatment of children, the parents’ working life was seriously endangered, as 46% were at risk of losing their job, 51% had limited job opportunities and 68% had to reduce their working hours [12]. Another study indicated a statistically significant 21% reduction in the mothers’ earnings and a statistically significant 10% reduction in the fathers’ earnings in the year of diagnosis compared to the control group of mothers and fathers, respectively [13].

From the very beginning of the diagnosis, the child’s disease has a strong influence on the physical, social and psychological dimensions of their caregivers. As numerous reports show, parents most often react to the diagnosis with shock, denial and a decreased life quality [14,15]. In the literature on parental mental health, increased levels of mental stress, such as anxiety, depression, sleep disturbances, somatic symptoms, fear of relapse, extensive worry and fatigue, were reported [16,17]. Studies have shown that 15% of parents are so far unable to come to terms with their child’s disease, 77% of them feel anxious and 11% take antidepressants. Overall, 53% of mothers of sick children feel exhausted, and the most common reported somatic problem was fatigue and difficulty sleeping. As reported by Beheshtipour et al., symptoms of parental burnout related to emotional exhaustion as a result of prolonged and severe stress may appear as early as 6 months after diagnosis [18]. Numerous publications by Kazak et al. show that the identification of family risk groups and the occurrence of psychosocial problems in a screening method allows for planning appropriate interventions of the oncological team in relation to the entire family and thus prevent unsolvable situations [19,20,21,22]. In addition, routine screening for psychosocial difficulties based on parental reports can be effective in increasing communication with healthcare professionals about psychosocial needs, as well as facilitating the selection of appropriate mental health services [23]. After discontinuing the dysfunction assessment scale due to existing biopsychosocial problems, this study showed that 34% of families were at the minimum risk level, 46% at the average level, and 20% at the high-risk level of psychosocial problems. Kazak et al. demonstrated that the majority of families (72%) were at the universal level, 24% were within the target range and only 2.4% were within the clinical range [19,20,21,22]. In a study by Gilleland et al., 51% of families in the “universal” category, 34% in the “target” category and 15% in the “clinical” category were included in the sample [23].

As parents, siblings of children with cancer may show symptoms and negative emotions because they also have to cope with the changed everyday life of the family and reduced physical and emotional availability of their parents [24,25,26,27]. Apart from the obvious support shown by siblings, there are conflicts and communication problems, as our research showed. In addition, this study showed that the siblings developed tantrums (48%), learning difficulties (21%) and behavioural problems (18%). Similar results were obtained by Rajajee et al., showing that siblings of a child suffering from cancer were also affected by both behavioural problems and academic performance [28].

At the time of diagnosis, the child’s disease inevitably affects family relationships. According to numerous reports, the changes in relationships occurred within weeks to 4 months after the child was diagnosed. When the child was ill for up to a year, the parents reported few changes in their relationships; from 2 to 3 years, they observed positive changes in their relationships; and from 4 years or more, the parents noticed slight changes [29,30,31]. In this research, only 27% of respondents had family relations deteriorate and the same number of people reported having conflicts in the family, but no relationship with the time of the child’s illness was found. Reported problems correlated with data from the literature and they were most often: conflicts due to the deterioration of intimacy, sexuality, communication and a lack of time for the family [30,31,32]. According to the report of Pai et al., mothers of children with cancer more often struggled with family conflicts and family disputes were related to the child’s disease [33]. Research by Colletti et al. showed that the source of conflicts was paying excessive attention to the sick child and the lack of attention towards the second offspring [34].

The survey showed that parents struggled with various problems related to care, somatic problems (55%) and psychological and social problems (61%). What makes it most difficult for parents to take care of a child is the lack of state support (57%). No studies on caring problems were found in the literature, except for reports by Reisi-Dehkordi et al., who drew attention to the loneliness of mothers in caring for a sick child, the lack of trust in nursing staff and the constant presence in the hospital [14].

This qualitative research covers many aspects of family functioning and experiences related to a child’s cancer. The strength of the study is the confirmation of the results of the studies described in the literature, and the obtained results provide better insight into the analysis of the child’s parents’ problems in all areas of life. The main weaknesses of the study were the small sample size, which limited the possibility of generalisation, and the fact that the caregivers in this sample were mostly women, which may limit the generalisation of the results.

Due to the lack of such studies in Poland, we are currently in a new team examining the problems of parents of children with cancer, broken down into fathers and mothers, and taking into account the duration of the disease and treatment. We want current and future research to become important to staff working with families of children with cancer to know what to look for and decide what support system and interventions to use during care.

## 6. Conclusions

Parents of children suffering from cancer report numerous psychological, social and somatic problems. The identification of problems through screening should translate into specific interventions, thus creating support for the families of children with cancer. Promoting coping with difficult emotions and the ability to solve problems when a child is ill has a positive effect on the functioning of the family.

## Figures and Tables

**Figure 1 children-08-00592-f001:**
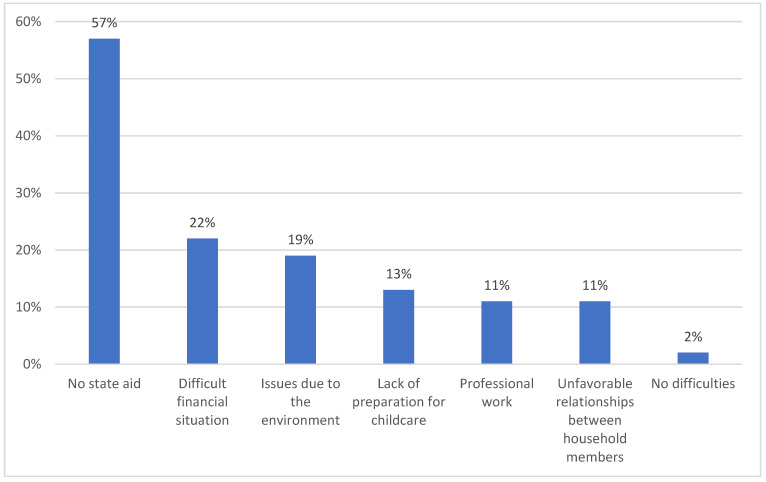
The sources of the caring problems.

**Figure 2 children-08-00592-f002:**
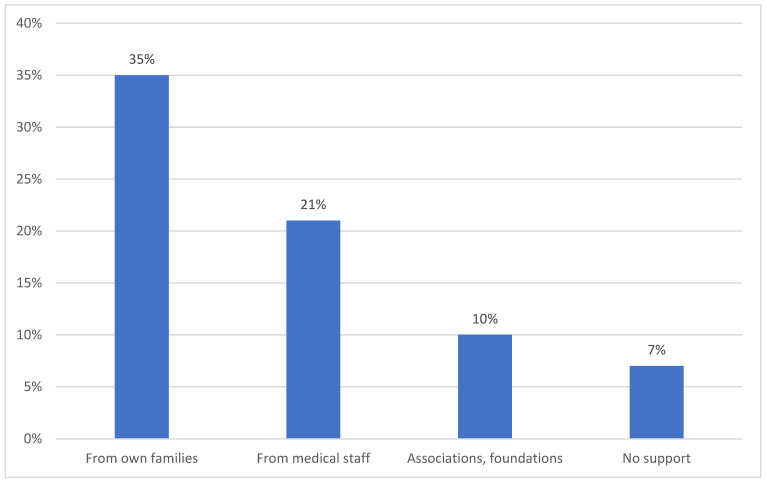
Sources of support.

**Figure 3 children-08-00592-f003:**
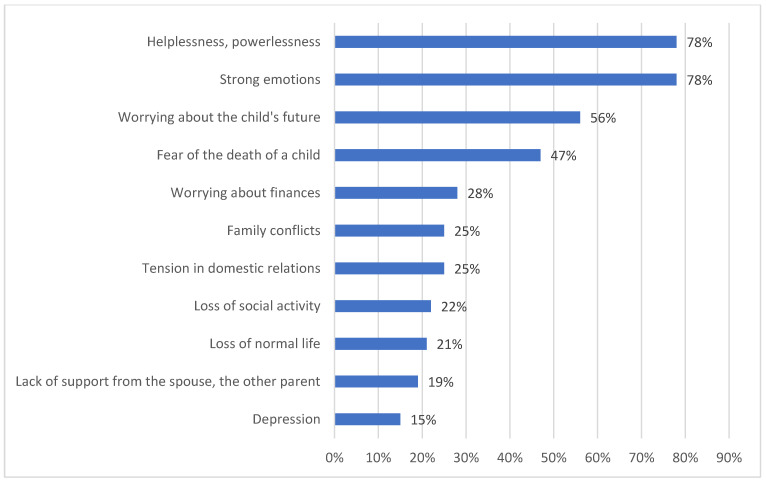
Reasons for using a psychologist.

**Figure 4 children-08-00592-f004:**
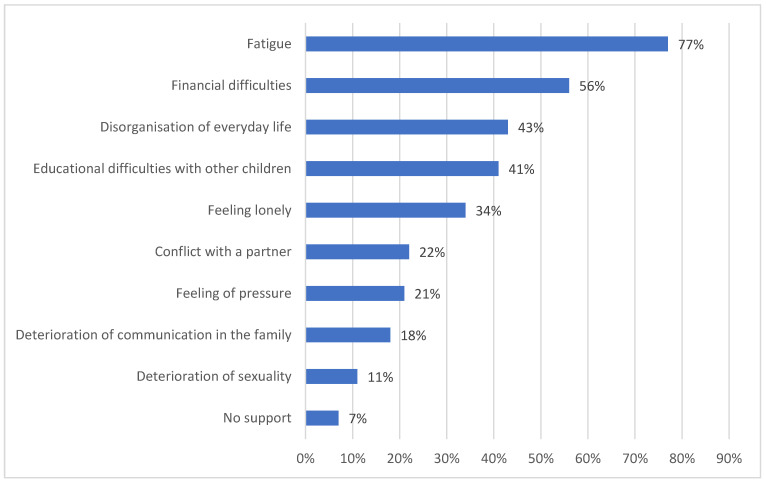
The source of family conflicts.

**Table 1 children-08-00592-t001:** Descriptive statistics of the examined group of patients.

Demographic Information	Total N = 800	*p*
Characteristics % (N)		
Sex		
Women	85% (680)	0.01
Men	15% (120)	
The age of the study group		
SD	44.1 (7.76)	0.12
95% CI	<26; 57>	
The age of women		
Age ± standard deviation	38.2 ± 7.25	0.21
Range	[26; 57]	
Median	38	
95% CI	[39.8; 41.8]	
The age of men		
Age ± standard deviation	41,1 ± 7,03	0.19
Range	[26; 57]	
Median	41	
95% CI	[39.8; 41.8]	
Place of residence		
City	68% (544)	0.21
Village	32% (256)	
Financial situation		
Very good	1% (8)	0.01
Good	8% (64)	
Average	68% (544)	
Bad	10% (80)	
Very bad	13% (104)	
Age groups		
20–29	3% (24)	0.01
30–40	35% (280)	
41–50	37% (296)	
51–60	25% (200)	
Education of the study group		
Higher education	47% (378)	0.01
Secondary education	35% (276)	
Vocational education	18% (146)	
Primary education	0% (0)	
Marital status		
Married	74% (592)	0.62
Widowed	3% (24)	
Unmarried	23% (184)	
Source of income		
Professionally active	76% (608)	0.59
Annuity	15% (120)	
Benefit	9% (72)	
Type of the child’s cancer		
Leukemia	54% (432)	0.07
Brain tumors	19% (152)	
Solid tumors	27% (216)	
Age of children with cancer		
Up to 5 years	22% (176)	0.19
5–10 years	51% (408)	
11–18 years	27% (216)	
Number of children in each family		
One child	45% (360)	0.71
Two children	41% (328)	
Three children	10% (80)	
Four children	4% (32)	
Duration of illness		
3–12 months	43% (344)	0.01
1–2 years	37% (296)	
3–4 years	20% (160)	

**Table 2 children-08-00592-t002:** Caring problems.

Area	Duration of the Disease			*p*	Place of Residence		*p*	Employment Status		*p*
	3–12 Months	1–2 Years	3–4 Years		City	Village		Working	Not Working	
Characteristics % (N)										
Somatic										
Limited independence	18% (68)	7% (26)	11% (41)	0.41	15% (56)	21% (79)	0.91	27% (101)	9% (68)	0.01
Disability	3% (11)	10% (38)	10% (38)	0.55	9% (34)	14% (53)	0.41	17% (64)	6% (68)	0.41
Skin problems	56% (210)	30% (113)	14% (53)	0.01	52% (195)	48% (180)	0.81	55% (207)	45% (68)	0.19
Gastric problems	46% (173)	26% (98)	7% (26)	0.01	38% (143)	41% (154)	0.77	39% (147)	40% (150)	0.88
Psychological										
Attention and memory deficits	14% (62)	25% (110)	18% (79)	0.41	24% (106)	33% (145)	0.91	21% (92)	36% (158)	0.64
Learning difficulties	11% (48)	23% (101)	20% (88)	0.71	28% (123)	26% (114)	0.55	30% (132)	24% (106)	0.88
Anxiety, restlessness	34% (150)	29% (128)	17% (75)	0.88	44% (194)	36% (158)	0.44	41% (180)	39% (172)	0.55
Depression	4% (18)	15% (66)	18% (79)	0.55	17% (75)	20% (88)	0.55	15% (66)	22% (97)	0.71
Mood swings	34% (150)	15% (66)	18% (79)	0.01	34% (150)	33% (145)	0.91	44% (194)	23% (101)	0.01
Social										
Difficulties in peer relationships	12% (58)	19% (93)	24% (117)	0.88	14% (68)	41% (200)	0.01	31% (151)	24% (117)	0.55
Sibling relationship problems	12% (58)	10% (49)	24% (117)	0.41	21% (102)	25% (122)	0.62	40% (195)	4% (19)	0.01
Reluctance to attend school	10% (49)	25% (122)	21% (102)	0.55	14% (68)	42% (205)	0.01	30% (146)	26% (127)	0.17
Insulation	13% (63)	18% (89)	27% (132)	0.59	16% (78)	42% (205)	0.01	32% (156)	26% (127)	0.19

**Table 3 children-08-00592-t003:** Symptoms checklist.

Symptoms	Duration of Illness			*p*	Sex		*p*
	3–12 Months	1–2 Years	3–4 Years		Women	Men	
Characteristics % (N)							
Fatigue	19% (65)	10% (30)	39% (62)	0.01	39% (265)	29% (35)	0.02
Weight loss	7% (24)	4% (12)	13% (21)	0.54	20% (136)	4% (5)	0.01
Loss of appetite	14% (48)	18% (53)	7% (11)	0.03	19% (129)	20% (24)	0.88
Gastric disorders	24% (82)	14% (41)	9% (14)	0.03	37% (252)	10% (12)	0.01
Headaches	22% (76)	11% (32)	9% (14)	0.88	20% (136)	22% (26)	0.91
Difficulty sleeping	19% (65)	4% (12)	28% (45)	0.01	29% (197)	22% (26)	0.71
Difficulty concentrating	9% (31)	11% (32)	19% (30)	0.09	19% (129)	20% (24)	0.74
Fear for the future	29% (100)	10% (30)	10% (16)	0.55	30% (204)	19% (23)	0.01

**Table 4 children-08-00592-t004:** Assessment of functioning.

Problems	Range			*p*
	0–10	11–20	21–30	
Characteristics % (N)				
Financial difficulties	8% (64)	10% (80)	5% (40)	0.71
Lack of social support	3% (24)	3% (24)	0% (0)	0.54
Lack of family support	2% (16)	2% (16)	0% (0)	0.03
Psychological problems	4% (32)	4% (32)	2% (16)	0.03
Problems in the relationship between partners	5% (40)	5% (40)	2% (16)	0.88
Problems with stimulants	4% (32)	4% (32)	1% (8)	0.19
Health problems	4% (32)	7% (56)	5% (40)	0.09
Problems with caring for a sick child	2% (16)	7% (56)	3% (24)	0.55
Problems with caring for siblings	2% (16)	4% (32)	2% (16)	0.19

## Data Availability

Data available on request due to restrictions of privacy and ethical.

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
