# Peer review of "Influence of a Child’s Cancer on the Functioning of Their Family"

_children, 2021, doi:10.3390/children8070592_

Round 1
Reviewer 1 Report
The topic of this study on parentl’s experiences on havinga child with cancer is interesting also for the adoptation of the qualitative approach. However the paper has several major limitations.
The introduction is poor and it doesn’t describe in detail the literature on this topic. I suggest to add some quotations and to discuss more them to introduce better the aims of the study.
For example you could add some sentences as this” Mothers could also develop post traumatic symptoms that could last for all the first year of therpies. Quote this: Tremolada M., Bonichini S., Schiavo S., Pillon M. (2012). Post-traumatic stress symptoms in mothers of children with leukaemia undergoing the first 12 months of therapy: Predictive models, Psychology & Health, 27(12):1448-62. DOI:10.1080/08870446.2012.690414
Also the aims of the study and the possible research questions were not explained in detail. It is not clear the value added of the study.
The method should contain a full description of instruments, also the Cronbach alpha values in this sample, bat also the cut off scores (if present), the Likert scale for each instrument and so on. The qualitative approach should be explained more, explained the value of this approach.
In the table 1 it is not clear the type of cancer diagnosis, it seems not English…and the same thing for the age of the children.
The values of the tests adopted in the analyses were not reported, for example at the end of paragraph 4.2 and at the end of paragraph 4.3 and in the other paragraphs. It is necessary to report the type of test run, the value of the test, the significance, the mean ranks or other parameters provided by each test.
In the discussion or in the conclusion it should be taken into consideration possible recommendations for future research, clinical suggestions for health professionals.
Author Response
Dear Reviewer,
I would like to thank the reviewers for their comments. After analysing all the comments, I made the following changes:
The introduction is poor and it doesn’t describe in detail the literature on this topic. I suggest to add some quotations and to discuss more them to introduce better the aims of the study. For example you could add some sentences as this” Mothers could also develop post traumatic symptoms that could last for all the first year of therpies. Quote this: Tremolada M., Bonichini S., Schiavo S., Pillon M. (2012). Post-traumatic stress symptoms in mothers of children with leukaemia undergoing the first 12 months of therapy: Predictive models, Psychology & Health, 27(12):1448-62. DOI:10.1080/08870446.2012.690414
- The introduction was expanded, more information and quotations were added, including those indicated by the Reviewer (line 43-81, 415-416).
Also the aims of the study and the possible research questions were not explained in detail. It is not clear the value added of the study.
- The objectives of the study are explained in more detail (line 82-87).
The method should contain a full description of instruments, also the Cronbach alpha values in this sample, bat also the cut off scores (if present), the Likert scale for each instrument and so on. The qualitative approach should be explained more, explained the value of this approach.
- The methodology describes the research methods, sample and statistics in more detail (line 90-178).
In the table 1 it is not clear the type of cancer diagnosis, it seems not English…and the same thing for the age of the children.
- Corrected to English (line 202-203).
The values of the tests adopted in the analyses were not reported, for example at the end of paragraph 4.2 and at the end of paragraph 4.3 and in the other paragraphs. It is necessary to report the type of test run, the value of the test, the significance, the mean ranks or other parameters provided by each test.
- More statistical details have been added (line 158-201).
In the discussion or in the conclusion it should be taken into consideration possible recommendations for future research, clinical suggestions for health professionals.
- Recommendations and suggestions have been added at the end of the discussion (line 389-393).
I hope that the changes made are satisfactory and this will allow publication. I am asking you to take into account the positive comments of the reviewers that this is an interesting study and a good study.
Sincerely
Anna Lewandowska

Reviewer 2 Report
This article is an interesting study to present the influence of a child's cancer on the functioning of the family in. Poland, and identified problems from psychological, social somatic aspects. However, I had some comments for present manuscript as below.
C1: The authors may describe more on why is this issue important. What can patients benefit from knowing the influence of a child's cancer on the functioning of the family?
C2: The data was restricted to only one single hospital in Poland, which may lack generalizability. The authors may describe more on the information of this Clinical Hospital. For example, is it an children-specific hospital? How many ill children are serviced per year in this hospital? It is a public hospital or private hospital? What’s the similar analysis results of other hospitals or countries so far? Besides, what can other countries learn from Poland experience?
C3: Participants were invited via outpatient visit setting or during child’s hospitalization, and complete the questionnaire. Is there any selection bias? For example, for the same child, father and mother may feel differently, how do we decide to use the opinion of father or mother? Also, some family may choose high care quality hospital to receive treatment even the cost is expensive, while some may not have any ability to go to any hospital. The author must state more on the composition of the study population.
C4: Why all the results of P-value are presented “, comma” but not “. period”? Besides, the design of questionnaire is composed of “financial problems”, “caring problems”, “psychological problems”, “somatic problems”, “communication problems”, “family problems”, “occurrence of dysfunction”. What is the reference of these items? Is it possible that we missed some domain that us not in above domain? Also, are these domains equally important or should be calculated by different weight?
Author Response
Dear Reviewer,
I would like to thank the reviewers for their comments. After analysing all the comments, I made the following changes:
C1: The authors may describe more on why is this issue important. What can patients benefit from knowing the influence of a child's cancer on the functioning of the family?
- The introduction was expanded, more information and quotations were added (line 43-81).
C2: The data was restricted to only one single hospital in Poland, which may lack generalizability. The authors may describe more on the information of this Clinical Hospital. For example, is it an children-specific hospital? How many ill children are serviced per year in this hospital? It is a public hospital or private hospital? What’s the similar analysis results of other hospitals or countries so far? Besides, what can other countries learn from Poland experience?
- The methodology describes the research methods, sample and statistics in more detail (line 90-178).
- So far, no similar research has been conducted in Poland.
C3: Participants were invited via outpatient visit setting or during child’s hospitalization, and complete the questionnaire. Is there any selection bias? For example, for the same child, father and mother may feel differently, how do we decide to use the opinion of father or mother? Also, some family may choose high care quality hospital to receive treatment even the cost is expensive, while some may not have any ability to go to any hospital. The author must state more on the composition of the study population.
- It is explained in more detail in the methodology. In future research, the group of fathers and mothers will be separated and the sample size will be statistically significant (line 90-178).
- In Poland, there is a public care system for oncological treatment.
C4: Why all the results of P-value are presented “, comma” but not “. period”? Besides, the design of questionnaire is composed of “financial problems”, “caring problems”, “psychological problems”, “somatic problems”, “communication problems”, “family problems”, “occurrence of dysfunction”. What is the reference of these items? Is it possible that we missed some domain that us not in above domain? Also, are these domains equally important or should be calculated by different weight?
- The entry in the questionnaire and the values of P (line 120-129, table 1,2,3,4).
I hope that the changes made are satisfactory and this will allow publication. I am asking you to take into account the positive comments of the reviewers that this is an interesting study and a good study.
Sincerely
Anna Lewandowska

Round 2
Reviewer 1 Report
The paper is really ameliorated and it runs very well now. Good job ancd compliments for your hard work.
best wishes
Reviewer 2 Report
Thanks for the author's explanation and supplement. All my comments were responded reasonably in detail, and the manuscript was also revised accordingly. I have no further comments.